# Neurotensin Gene rs2234762 C>G Variant Associates with Reduced Circulating Pro-NT Levels and Predicts Lower Insulin Resistance in Overweight/Obese Children

**DOI:** 10.3390/ijms24076460

**Published:** 2023-03-30

**Authors:** Federica Sentinelli, Ilaria Barchetta, Flavia Agata Cimini, Sara Dule, Diego Bailetti, Efisio Cossu, Arcangelo Barbonetti, Maria Totaro, Olle Melander, Maria Gisella Cavallo, Marco Giorgio Baroni

**Affiliations:** 1Department of Clinical Medicine, Public Health, Life and Environmental Sciences (MeSVA), University of L’Aquila, 67100 L’Aquila, Italy; 2Department of Experimental Medicine, Sapienza University of Rome, 00161 Rome, Italy; 3Department of Medical Sciences and Public Health, University of Cagliari, 09042 Monserrato, Italy; 4Department of Clinical Sciences Malmö, Lund University, 20213 Malmö, Sweden; 5Department of Internal Medicine, Skåne University Hospital, 20213 Malmö, Sweden; 6Neuroendocrinology and Metabolic Diseases, IRCCS Neuromed, 86077 Pozzilli, Italy

**Keywords:** neurotensin, single nucleotide polymorphism, gene, obesity, children, insulin resistance, gastrointestinal peptides, blood lipids

## Abstract

Neurotensin (NT) is a small protein implicated in the regulation of energy balance which acts as both a neurotransmitter in the central nervous system and as a gastrointestinal peptide. In the gut, NT is secreted after fat ingestion and promotes the absorption of fatty acids. The circulating levels of its precursor, pro-NT, predicts the presence and development of metabolic and cardiovascular diseases. Despite the extensive knowledge on the dynamic changes that occur to pro-NT = after fat load, the determinants of fasting pro-NT are unknown. The aim of this study was to determine the possible genetic regulation of plasma pro-NT. The NT gene (NTS) was sequenced for potential functional variants, evaluating its entire genomic and potentially regulatory regions, in DNA from 28 individuals, stratified by low and high pro-NT levels. The identified variant differently distributed in the two pro-NT subgroups was genotyped in a cohort of nine hundred and thirty-two overweight/obese children and adolescents. A total of seven sequence variations across the NTS gene, none of them located in coding regions, were identified. The rs2234762 polymorphism, sited in the NTS gene promoter, was statistically more frequent in the lowest pro-NTS level group. Carriers of the rs2234762 variant showed lower pro-NT levels, after adjusting for sex, age, BMI, triglycerides and the Tanner stage. Having NTS rs2234762 predicted less pronounced insulin resistance at the 6.5-year follow-up with OR: 0.46 (0.216–0.983), at the logistic regression analysis adjusted for age, sex and BMI. In conclusion, the NTS rs2234762 gene variant is a determinant of reduced circulating pro-NT levels in overweight and obese children, which predisposes this group to a more favorable metabolic profile and a reduced insulin resistance later in life, independently from metabolic confounders.

## 1. Introduction

Neurotensin (NT) is a 13 amino acid peptide first identified as a neuropeptide in the central nervous system [1] where it mediates circuits associated with the regulation of energy balance, appetite and physical activity, as a primary neurotransmitter in dopaminergic areas [2]. Years later, NT was demonstrated to be secreted by the enteroendocrine N cells of the small intestine mucosa, where it is released after fat ingestion. Once secreted, NT regulates intestinal lipid absorption by binding three specific receptors named neurotensin receptor-1 (NTSR1), -2 (NTSR2) and -3 (NTSR3) which are differently distributed in diverse brain areas and visceral organs [3,4]. The effects of NT largely depend on the receptor type and distribution within tissues and organs. The release of NT from gut neuroendocrine cells is stimulated by the intestinal fatty acids’ concentration. Once secreted, it promotes lipid absorption through gut mucosa and regulates the secretion of other gastrointestinal peptides via endocrine and paracrine circuits [5].

Experimental studies have demonstrated that circulating concentrations of pro-NT, the stable NT precursor fragment released in equimolar amounts relative to NT, rise in response to the ingestion of high-fat food and high concentrations of intestinal fatty acids [6,7,8], along with bile acids and the concentration of triglycerides in the bloodstream. Neurotensin promotes intestinal lipid absorption, and a parallel pro-NT and triglyceride increase has been shown during both jejunal infusion of short-, medium-, and long-chain fatty acids, (with a peak after 60 min from infusion [6,7]), and after an oral lipid load in healthy subjects [8]. Moreover, in conditions of chronically high caloric excess, the fat intake and high pro-NT release induce the expression of the specific receptor NTSR1 at the gut level, thus facilitating the lipid influx through the intestinal mucosa [9]. Therefore, as a result, a diet rich in fat increases pro-NT, which, in turn, enhances the activity of gut NTSR1, resulting in augmented lipid absorption, visceral fat accumulation and an increase in body weight [9,10].

In longitudinal investigations, elevated fasting pro-NT levels predicted the presence and development of metabolic diseases such as obesity [10,11], insulin resistance [12,13], type 2 diabetes [11,14] and non-alcoholic fatty liver disease (NAFLD) [15]. High pro-NT is also an independent risk factor for cardiovascular diseases and overall mortality in several cohorts [11,16,17].

Recently, we demonstrated that elevated plasma pro-NT levels in a population of overweight/obese children and adolescents were significantly associated with bodyweight increase and impaired beta-cell secretion later in life; therefore, circulating pro-NT may be considered a marker of susceptibility to metabolic alterations in the presence of obesity in childhood [18].

The human neurotensin gene (NTS) maps on chromosome 12 (q21.31) spanning 8689 bps of genomic sequence [19] and is divided into 4 exons by 3 introns. The fourth exon encodes a common precursor for two peptides: neurotensin (NT) and neuromedin (NMN). The pro-NT/NMN consists of a conserved polypeptide of 170 aa residues starting with a signal peptide of 23 aa residues and are flanked and separated by 3 Lysine-Arginine sequences (KR). These KR sequences are consensus sites that are recognized and cleaved by specialized endoproteases [20].

Gut-derived NT accounts for the overall pro-NT levels detected in the bloodstream. Systemic pro-NT was demonstrated to cross the brain–blood barrier, so influencing NT-mediated brain circuits associated with appetite and energy balance regulation. Therefore, in physiological conditions, the rise in NT after fat-rich meals promotes satiety [21,22]; however, in obesity, chronically high pro-NT levels and the over-stimulation of the NT/NTSRs axis act to downregulate the brain’s NT pro-anorexigenic activity [23], promoting a positive energy balance and the development of metabolic complications of obesity [24].

Indeed, the content of fatty acids in the gut, which induces an NT secretion and a plasma pro-NT peak between two to three hours after fat ingestion, demonstrates that the fat content of food is a determinant of the acute pro-NT release into the bloodstream. However, the determinants of fasting pro-NT levels have not yet been identified.

A powerful strategy to discover causal variants involved in modulating a phenotype in association studies is to perform genetic studies in samples from the extremes of a quantitative trait [25]. Intuitively, trait-influencing alleles will be enriched in frequency in such samples, thus improving statistical power to discover risk variants and to detect their association to the trait [26,27,28]. Additionally, this strategy ensures more homogeneity of the cohorts under study and a greater power in detecting genetic associations and genetic markers with higher odds ratios (OR) [29].

To determine the possible genetic regulation of pro-NT, we designed a sequencing study to explore the existence of genetic variants associated with circulating pro-NT levels. We also investigated the metabolic phenotype of study participants at baseline and after a 6.5 year follow-up in relation to NTS gene variants.

## 2. Results

### 2.1. NTS Gene Variants and Pro-NT Concentrations

We designed a two-step genetic association study with the aims of exploring (i) the NTS gene for potential functional variants in its entire genomic and potential regulatory regions, and (ii) the possible association of NTS variants with plasma pro-NT levels in relation to the presence of obesity in children.

Step 1 consisted of an initial screening of 28 individuals, stratified by low and high pro-NT levels and divided into 2 equal groups, (as described in the Section 4). In step 2, we genotyped 1 sequence variation which was differentially distributed in the 2 pro-NT subgroups of 932 children with obesity. Here we were looking for an association with obesity and obesity-related traits.

### 2.2. Screening for Sequence Variations in NTS Gene

We identified a total of seven sequence variations, all reported in the public SNP database, across the NTS gene, none of which were located in the coding regions (Figure 1). A total of four polymorphisms, (rs2234762 C>G, rs1800832 A>G, rs546594364 G>A, rs560139347 C>G), were found in the regulatory/5′UTR region of the gene; the rs58553548 A>T variant was located in intron 3, and two polymorphisms (rs11117072 G>A, rs139226362 delGATT) were located in the 3′UTR region.

Both the rs2234762 and rs58553548 variants were statistically more frequent in the lowest pro-NTS level group (*p* = 0.029 and *p* = 0.012, respectively, χ^2^ test applied) and showed very strong linkage disequilibrium (D’ = 0.969).

The rs546594364 and the rs560139347 polymorphisms were present in only one subject. The rs1800832, rs11117072 and rs139226362 variants were present in both the low and the high pro-NT level group with similar frequencies.

### 2.3. NTS rs2234762 Variant and Metabolic Phenotype at the Baseline

The rs2234762 variant was selected between the two differentially distributed SNPs in the two pro-NT subgroups for its remarkable position in the regulatory region of the NTS gene as it is close to a CRE/AP-1-like element that is crucial for constitutive NTS gene expression [30]. It was genotyped in each of the 932 study participants.

The G allele frequency of the rs2234762 variant was 0.2, which was comparable to the allele frequency of 0.25 for the European (non-Finnish) population reported in gnomAD (https://gnomad.broadinstitute.org, accessed on 6 May 2021). Genotypes and allele frequencies did not deviate from the Hardy–Weinberg equilibrium taking into account our value of χ^2^ test = 0.65 which is below the critical value of 3.84 with 1 degree of freedom (*p* > 0.05).

Assuming a dominant mode of inheritance, GG homozygotes (33 in our sample) and CG genotypes were combined and compared to CC homozygotes.

In contrast to the wild type, carriers of the rs2234762 variant showed significant reduced levels of pro-NT levels (31.6 ± 20.8 vs. 22.2 ± 13.5 pmol/L, *p* = 0.007) and this association persisted after adjusting for potential confounders, such as sex; age; BMI; triglycerides and the Tanner stage (standardized β = −0.24, *p* = 0.003; Table 1). Bivariate analyses between plasma pro-NT and clinical/biochemical parameters are shown in Appendix A.

Children carrying the NTS rs2234762 variant also had lower LDL-cholesterol (104.3 ± 28.8 vs. 100.6 ± 26.6 mg/dL, *p* = 0.043), FBI (15.4 ± 9.5 vs. 14 ± 8.5 μUI/mL, *p* = 0.036) and less pronounced insulin resistance (HOMA-IR: 3.4 ± 2.2 vs. 3.1 ± 2, *p* = 0.033) than those in the non-carrier subgroup. Moreover, the NTS rs2234762 variant frequency was significantly reduced in obese children during the χ^2^ test (*p* = 0.041). Clinical characteristics of the study subjects, stratified by genotypes, are shown in Table 2.

Finally, the NTS rs2234762 variant was associated with the presence of a more favorable metabolic profile in overweight/obese children, as demonstrated by the multivariate linear regression model showing an independent relationship between the rs2234762 variant and lower HOMA-IR, after adjusting for age, sex, BMI and the Tanner stage (standardized β: −0.063, *p* = 0.041; Table 3).

### 2.4. NTS Gene Variant and Metabolic Phenotype at the Follow-Up Evaluation

All children who took part in the investigation at baseline were recalled, and 201 (91 males and 110 females, mean age 16.2 ± 3.5 and 16.6 ± 3.8 years, respectively) of them accepted to participate in the follow-up examination after a median 6.5 (range 3.5–10) years.

At the follow-up visit, anthropometric parameters were assessed and biochemistry and metabolic profiling were conducted, including OGTT execution and pro-NT measurement.

Within the participants undergoing follow-up examination, seven were homozygous and sixty-one heterozygous for the rs2234762 variant. Assuming a dominant mode of inheritance, GG homozygotes and CG genotypes were combined and compared to CC homozygotes.

Circulating pro-NT levels significantly increased throughout the lifespan (28.2 ± 19.3 vs. 92.8 ± 49.8 pmol/L, *p* < 0.001), as previously described by our group [18]. No association was found between the NTS rs2234762 variant and pro-NT levels measured at the follow-up; similarly, no difference was reported in the overall comparison between carriers and non-carriers in terms of metabolic parameters, although slightly reduced FSI (13.6 ± 8.3 vs. 15.7 ± 9.4 μUI/mL), HOMA-IR (2.9 ± 2 vs. 3.4 ± 2) and HOMA-β (228.5 ± 165.8 vs. 264 ± 216.8) were reported in individuals with the G allele.

It was noteworthy that the NTS rs2234762 variant was again associated with less pronounced insulin resistance, i.e., at the logistic regression analysis, the NTS rs2234762 variant associated with lower HOMA-IR distribution at the 6.5 year follow-up (HOMA-IR index below the median value 2.84) after adjusting for age, sex and BMI with OR: 0.46 (95% CI 0.216–0.983) (Table 4).

## 3. Discussion

In the present study, we identified two NTS gene variants never previously studied, significantly associated with pro-NT levels. Here, we demonstrate that the NTS rs2234762 variant is associated in children with lower circulating pro-NT concentration and that this correlation persists is highly significant after correcting for potential confounding parameters such as age, sex, the Tanner stage and body mass index.

By re-sequencing the NTS gene in individuals belonging to the extreme tails of plasma pro-NT distribution, we identified seven polymorphisms with no evidence of non-synonymous variations, among them, the rs2234762—located in the regulatory/5′UTR region of the gene—and the rs58553548—located in the intron 3—these variants were associated with reduced circulating pro-NT levels. Therefore, for the purposes of this study, we adopted the strategy of phenotypic extremes selecting subgroups of individuals within a homogeneous population who only differ in the test variable, i.e., pro-NT concentration. This could increase the likelihood of identifying genetic factors capable of influencing the test parameter [25,26,27,28].

In large population studies, fasting pro-NT does not associate cross-sectionally with raw indicators of total body mass, per se, in either normal weight, or in overweight or obese individuals [11,13], nor in adults or children [18]. Thus, the increase in fat mass does not seem to entirely determine differential pro-NT levels in obesity.

In this study, we hypothesized that genetic variants of the NT gene could, at least partially, explain differential pro-NT levels found in obese individuals. Therefore, we genotyped the rs2234762 variant in the entire study population, which includes obese participants recruited during childhood when the potential confounding effect of environmental factors associated with pro-NT, i.e., lifestyle, cardiovascular risk factors, and overt cardio-metabolic disease [11,16,17], could be less pronounced.

We observed that rs2234762 was associated with the presence of a more favorable metabolic profile in the study population and lower HOMA-IR. At follow-up, the NTS rs2234762 variant was associated with a marked reduction in insulin resistance, which confirmed the protective effect of the polymorphism.

It could be speculated that rs2234762 genetic variant may influence protein synthesis since it is located in the promoter of the NTS gene. Its location is close to a region, which contains a CRE/AP-1-like element crucial for the binding of both AP-1 and CREB/ATF proteins and for constitutive NTS gene expression [30].

To our knowledge, this is the first study to highlight the role of the rs2234762 variant in modulating the circulating pro-NT levels. In vitro studies may definitively provide mechanistic insights on the role of this NTS gene variant in regulating not only pro-NT production and secretion, but also in influencing clinical phenotype and cardio-metabolic risk.

With regards to other NTS variants, previous investigations observed that the rs1800832 polymorphism in the 5′UTR of the NTS gene was associated with pro-NT circulating levels [31,32,33], however, the results were discordant. Specifically, the studies performed by Riezzo and colleagues [31] and by Russo and colleagues [32], reported that the rs1800832 polymorphism was associated with reduced pro-NT levels. Conversely, Dongiovanni and colleagues [33] observed that pro-NT circulating levels were higher in patients with advanced fibrosis and hepatocellular carcinoma and even more so in carriers of the rs1800832 G allele. Unlike what was observed in these investigations, the rs1800832 variant was not associated with circulating pro-NT levels in our study population, as the polymorphism was equally distributed in both the low and the high pro-NT level groups.

The strength of this study is the investigation performed in children, a model that offers a reduced effect of confounding factors, such as dietary behavior, physical activity and environmental conditions, on the trait under study, and increases the possibility that the genetic predisposition will become more clinically evident.

One possible limitation is that our genetic study is population specific as only Caucasian children and adolescents were studied and the rs2234762 genetic variant could influence the pro-NT levels in other ethnic populations differently. Moreover, since the NTS rs2234762 C>G gene variant is associated with a more favorable metabolic phenotype, studies in normal weight populations could unravel a differential in the prevalence of the NTS rs2234762 C>G gene variant, where it might contribute to determining a metabolically favorable genotype in the presence of traditional/environmental risk factors for obesity and metabolic disease.

## 4. Materials and Methods

### 4.1. Study Population

A total of 932 overweight/obese children and adolescents (M/F: 441/491; mean ± SD age 10.4 ± 3.2 years) attending the outpatient clinic of the Paediatric Endocrine Unit of the Regional Hospital for Microcitaemia; Cagliari; Italy, were consecutively recruited. Enrolment started in May 2007 and ended in May 2010. After a median 6.5 (range 3.5–10) years, all participants were recalled, and 201 (91 males and 110 females) of them agreed to participate in the follow-up examination which was performed between 2013 and 2016.

In all cases, the reason for a medical referral was the presence of body weight excess. All children were instructed to follow an educational program that involved dietary and physical activity modifications. The exclusion criteria were the presence of endocrine disorders or genetic syndromes which included syndromic obesity and chronic pharmaceutical treatments.

### 4.2. Clinical Evaluations and Laboratory Assessment

All subjects underwent a complete clinical work-up including physical examination, anthropometric measurements and laboratory tests for metabolic profiling. Serum samples were collected after an overnight fast, then they were aliquoted and stored at −80 °C until experiments were performed.

Systolic and diastolic blood pressure was measured 3 times after a 10 min rest and the average of these measurements was used in the analysis.

Overweight and obesity, as well as standard deviation score body mass index (SDS–BMI), were defined according to the validated Italian growth charts for people aged 2–20 years [34]. The pubertal developmental stage was determined using the Tanner definition [35]; participants were divided accordingly in two subgroups: the pre-pubertal (Tanner stage I: boys with pubic hair and gonadal stage I, girls with pubic hair stage and breast stage I) and pubertal one (Tanner stages ≥ II–V: boys with pubic hair and gonadal stage ≥ II and girls with pubic hair stage and breast stage ≥ II).

The study protocol was reviewed and approved by the University of Cagliari Ethical Committee (ref n. 45/08/CE) and conducted in conformance with the Helsinki Declaration. Informed written consent was obtained from the participants or their legal guardians.

Blood samples were obtained from the antero-cubital vein after 12 h fasting for evaluating routine biochemistry and metabolic profile, including plasma glucose (FBG, mg/dL) and insulin (FSI, IU/mL) at 0 and 120 min, total cholesterol (mg/dL), high-density lipoprotein cholesterol (HDL-C, mg/dL), triglycerides (mg/dL), aspartate aminotransferase (AST, IU/L) and alanine aminotransferase (ALT, IU/L). Plasma glucose was determined by the glucose oxidase method (Autoanalyzer, Beckman Coulter, Brea CA, USA). Plasma insulin levels were measured with a radio immunoassay on samples which were stored at −80 °C (DLS-1600 Insulin Radioimmunoassay Kit; Diagnostic System Laboratories Inc., Webster, TX, USA). AST, ALT, total cholesterol HDL-C and triglycerides were measured in the local laboratory by standard methods.

Low-density lipoprotein cholesterol (LDL-C) value was calculated using the Friedewald formula.

The standard oral glucose tolerance test (OGTT) was performed according to clinical recommendations for children (1.75 g oral glucose administration per kg body weight, up to 75 g).

Systolic and diastolic blood pressure was measured 3 times after a 10 min rest and the average of these measurements was used in the analysis.

The homeostasis model assessment of insulin resistance (HOMA-IR) for insulin resistance assessment, HOMA-β for insulin secretion evaluation, the insulin-sensitivity index (ISI) and the single-point insulin sensitivity estimator (SPISE) index were calculated as previously shown by Matthews et al. [36], Matsuda et al. [37] and Barchetta I et al. [38].

### 4.3. DNA Sequence Variations and Genotyping of the NTS Gene

To explore the existence of genetic variants associated with differential plasma NT levels, we measured circulating concentrations of pro-NT, the stable NT precursor fragment released in equimolar amounts relative to NT, in 150 children representative of the entire study population. Pro-NT was assessed using a chemiluminometric sandwich immunoassay to detect pro-NT amino acids 1–117 as previously described [18].

Then, we selected subjects belonging to the 10th and 90th percentile for plasma pro-NT levels (pro-NT < 10.1 pmol/L versus pro-NT > 48.6 pmol/L) and common variants in the NTS gene were identified by direct resequencing.

For this purpose, primers for the 4 exons of the NTS gene, together with its 5′ regulatory region (up to −651 base pairs) and the 3′-UTR region, were newly designed with reference to the NTS genomic sequence ID ENST00000256010.7. Sequence of primers are available upon request.

Genotyping of the rs2234762 C>G SNP was assayed using the TaqMan assays (Applied Biosystems, Waltham MA, USA) C__11922676_10. The assay was carried out on an Eco™ Real-Time PCR System by Illumina (San Diego, CA, USA) in a total volume of 10 μL.

### 4.4. Statistics

Power calculation: the size of the study cohort had a power >80% to detect, with an alpha error of 0.05, a 9.4 pmol/L (29.7%) difference in pro-NT levels across different genotypes.

SPSS version 25.0 statistical package was used to perform the analyses. These included Student’s *t*-test and Mann–Whitney nonparametric independent sample test for mean comparison between normally and non-normally distributed variables, respectively, and χ^2^ test for categorical variables, as appropriate. Correlations were estimated by Spearman’s rho non-parametric test or Pearson’s correlation coefficient. The Wilcoxon’s rank test for paired samples was conducted to compare clinical and biochemical parameters of study subjects at baseline and after follow-up. Multivariate regression models were built in order to identify independent predictors of (i) pro-NT levels and (ii) HOMA-IR, as considered as continuous parameters, entering variables significantly associated at the bivariate analyses and forcing for sex/age. Furthermore, the odds ratio of the NTS gene variant rs2234762 in predicting the development of insulin resistance (estimated by HOMA-IR) later in life was calculated by a multivariate logistic model adjusted for age, sex and BMI at the time of the follow-up visit. For all analyses, *p*-values < 0.05 were considered statistically significant with a confidence interval of 95%.

## 5. Conclusions

In conclusion, this is the first report on the NTS rs2234762 gene variant as a determinant of reduced circulating pro-NT levels in overweight and obese children, which predisposes them to a more favorable metabolic profile and a reduced insulin resistance later in life, independently from metabolic confounders. Indeed, the NTS rs2234762 variant may confer to carriers a certain protection against metabolic consequences of chronic caloric excess, via lower pro-NT levels per similar obesity class. Finally, the study findings (which warrant confirmation in larger populations) add novel insights into the pathophysiological processes leading from fat accumulation to metabolic disease in humans and provide novel data on the genetic regulation of NT as a gastrointestinal peptide deeply implicated in insulin resistance and cardiovascular risk profiles.

## Figures and Tables

**Figure 1 ijms-24-06460-f001:**
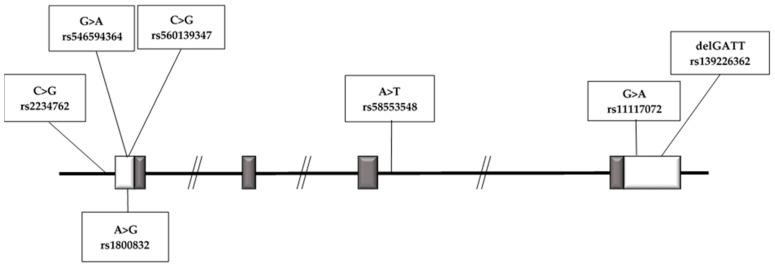
Neurotensin gene (NTS) identified polymorphisms.

**Table 1 ijms-24-06460-t001:** Multiple linear regression analysis between plasma pro-NT levels and rs2234762 variant.

	Non-Standardized β Coefficient	Standard Error	Standardized β Coefficient	*p*-Value
Age	2.648	0.883	0.441	0.003
BMI	−1.024	0.553	−0.216	0.066
Sex (M vs. F)	3.310	3.027	0.086	0.276
Tanner stage	−2.570	4.628	−0.067	0.580
Triglycerides	0.082	0.049	0.138	0.098
rs2234762 (yes vs. no)	−9.659	3.171	−0.239	0.003

Plasma pro-NT concentration, considered as continuous variable, is the dependent variable. *p* values < 0.05 are considered significant. Abbreviations: BMI, body mass index.

**Table 2 ijms-24-06460-t002:** Clinical characteristics of overweight and obese children and adolescents stratified by NTS genotype.

	Genotypes		*p*
	CC n = 594	CG+GG n = 338	
Sex (M vs. F)	328/266	163/175	0.039 *
Age (years)	10.4 ± 3.2	10.3 ± 3.2	0.769
Tanner stage 1/≥2	299/289	168/169 §	0.770
Pro-NT (pmol/L)	31.6 ± 20.8	22.2 ± 13.5	0.007
BMI (kg/m^2^)	27.6 ± 4.2	27.3 ± 4.8	0.327
SDS-BMI	2.1 ± 0.4	2 ± 0.4	0.062
SBP (mm/Hg)	106.2 ± 14.6	105.4 ± 14.9	0.398
DBP (mm/Hg)	62.2 ± 8.8	61 ± 8.7	0.046
TC (mg/dL)	168.3 ± 32.5	167.3 ± 32	0.193
TG (mg/dL)	63.1 ± 38.1	63.5 ± 41.4	0.881
HDL-C (mg/dL)	51.3 ± 12	51.6 ± 12.4	0.692
LDL-C (mg/dL)	104.3 ± 28.8	100.6 ± 26.6	0.043
AST (U/L)	24.7 ± 12	25.6 ± 5.9	0.318
ALT (U/L)	23.4 ± 11.7	22.4 ± 13.5	0.253
FBG 0′ (mg/dL)	89 ± 7.4	89 ± 8	0.899
FBG 120′ (mg/dL)	105.3 ± 17	106.5 ± 24.5	0.407
FSI 0′ (μUI/mL)	15.4 ± 9.5	14 ± 8.5	0.036
FSI 120′ (μUI/mL)	63.1 ± 53.5	60.7 ± 48.8	0.505
HOMA-IR	3.4 ± 2.2	3.1 ± 2	0.033
HOMA-β	223.8 ± 146	211.8 ± 233.6	0.037
ISI	6.7 ± 5.1	7.4 ± 6.4	0.068
SPISE	6.8 ± 1.6	6.9 ± 1.7	0.215

§ For 7 subjects, data are missing. Student’s *t*-test applied; * χ^2^ test applied. Abbreviations: CC, homozygous for C allele; CG, heterozygous carrying both C and G alleles; GG, homozygous for G allele; pro-NT, pro-neurotensin; BMI, body mass index; SDS BMI, standard deviation score of body mass index; SBP, systolic blood pressure; DBP, diastolic blood pressure; TC, total cholesterol; TG, triglycerides; HDL-C, high-density lipoprotein; LDL-C, low-density lipoprotein; AST, alanine aminotransferase; ALT, aspartate aminotransferase; FBG, fasting blood glucose; FSI, fasting serum insulin; HOMA-IR, homeostatic model assessment of insulin resistance; HOMA-β, homeostatic model assessment for β-cell function; ISI, insulin-sensitivity index; SPISE, single-point insulin sensitivity estimator. TANNER I = pre-pubertal, TANNER ≥ II = pubertal stage.

**Table 3 ijms-24-06460-t003:** Multiple linear regression analysis between HOMA-IR index and rs2234762 variant.

	Non-Standardized β Coefficient	Standard Error	Standardized β Coefficient	*p*-Value
Age	−0.023	0.030	−0.035	0.444
BMI	0.161	0.018	0.338	<0.001
Sex (M vs. F)	0.262	0.135	0.062	0.053
Tanner stage	0.694	0.178	0.164	<0.001
rs2234762 (yes vs. no)	−0.279	0.136	−0.063	0.041

HOMA-IR is the dependent variable. Abbreviations: BMI, body mass index.

**Table 4 ijms-24-06460-t004:** Logistic regression analysis with median HOMA-IR as a dependent variable.

Variable	OR	95% CI	*p*
Age	0.851	0.768–0.942	0.002
Gender	0.679	0.332–1.386	0.287
BMI	1.282	1.173–1.401	<0.001
rs2234762	0.461	0.216–0.983	0.045

Abbreviations: BMI, body mass index.

## Data Availability

The data presented in this study are available on reasonable request. The data are not publicly available due to privacy restrictions, lacking specific patients’ consent.

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
