# Peer review of "Neurotensin Gene rs2234762 C>G Variant Associates with Reduced Circulating Pro-NT Levels and Predicts Lower Insulin Resistance in Overweight/Obese Children"

_ijms, 2023, doi:10.3390/ijms24076460_

Round 1

Reviewer 1 Report

A minor spell check is suggested. See lines 172, and 340 of the manuscript. 

Title: Neurotensin gene rs2234762 C>G variant associates with reduced circulating pro-NT levels and predicts lower insulin-resistance in overweight/obese children
Summary
This article reports the identification of a gene variant as a biomarker for pro-NT levels and insulin resistance in overweight and obese children. Further investigations into the significance of the identified gene in normal and diseased pathways are warranted.
Comments to Authors
a. Would the addition of non-obese/overweight participants as a control population provided a broader sample set to evaluate the prevalence/function of this gene variant?
b. Do in vitro studies indicate that the rs2234762 C>G single nucleotide polymorphism influence pro-NT expression and subsequent release of NT?

Author Response

We do agree with the reviewer that further investigations into the significance of the NTS gene in normal and diseased pathways, in terms of both epidemiological and functional studies, are warranted. In this investigation, we explored genetic determinants of circulating pro-NT concentrations describing a novel association with the NTS rs2234762 C>G gene variant, which predicts lower pro-NT levels along with better metabolic profile in the long term, in overweight and obese individuals.

  1. In particular, since the NTS rs2234762 C>G gene variant is associated with more favourable metabolic phenotype, studies in normal weight populations could unravel differential prevalence of the NTS rs2234762 C>G gene variant, where it might contribute to determine some kind of “metabolically favourable genotype” also in presence of traditional/environmental risk factors for obesity and metabolic disease. Nonetheless, all the analyses performed in this study have been adjusted -among other confounders- for SDS-BMI, and confirmed that NTS rs2234762 C>G gene variant carriers had lower pro-NT levels and that this genotype associated with more favourable metabolic profile regardless of BMI.
  2. To our knowledge, this is the first study to highlight a role of the rs2234762 variant in modulating the circulating pro-NT levels, and there are no functional studies yet. In vitro studies may definitively provide mechanistic insights on the role of this NTS gene variant in regulating not only pro-NT production and secretion, but also in influencing clinical phenotype and cardio-metabolic risk.

We do thank the reviewer for these bright observations and have added a novel paragraph in the Discussion section commenting on them: “To our knowledge, this is the first study to highlight a role of the rs2234762 variant in modulating the circulating pro-NT levels. In vitro studies may definitively provide mechanistic insights on the role of this NTS gene variant in regulating not only pro-NT production and secretion, but also in influencing clinical phenotype and cardio-metabolic risk”. (Page 6, lines 48-51).

“Since the NTS rs2234762 C>G gene variant is associated with more favourable metabolic phenotype, studies in normal weight populations could unravel differential prevalence of the NTS rs2234762 C>G gene variant, where it might contribute to determine a “metabolically favourable” genotype also in presence of traditional/environmental risk factors for obesity and metabolic disease".  (Page 7, lines 17-22).

Reviewer 2 Report

The aim of the article by Sentinelli F. et al is to use genetic analysis to evaluate the possible genetic regulation of plasma pro-NT level, a precursor of neurotensin, continuous circulation of which is associated with development of many dangerous human diseases and obesity. The NT gene (NTS) was sequenced and 7 polymorphic variations were found in the non-coding regions and analyzed in two groups of children stratified by low and high pro-NT levels. Finally, a most promising variant rs2234762 was found located close to the interesting regulatory region at 5’ termini. The authors have convincingly shown that carriers of such variant rs2234762 have many advantages compared to wild type: reduced pro-NT level, lower LDL-cholesterol level, less obesity, less pronounced insulin resistance (HOMA-IR). Multiple linear regression analysis validates the data obtained adjusting for potential confounders (sex, age, BMI etc). The only factor reducing the significance of the work is that only Caucasian children were analyzed and the effect of rs2234762 variant might depend on the ethnic group. The authors agree that current study should be considered as a pilot one and genetic analysis of more wide population groups is necessary to confirm the significance of rs2234762 variant. In any case the data obtained is interesting and should be published after correction of the slip on line 340 “confiurmation” should be changed to confirmation.

Please see more comments in the attachment.

Author Response

Reviewer 2

Comments and Suggestions for Authors

 The aim of the article by Sentinelli F. et al is to use genetic analysis to evaluate the possible genetic regulation of plasma pro-NT level, a precursor of neurotensin, continuous circulation of which is associated with development of many dangerous human diseases and obesity. The NT gene (NTS) was sequenced and 7 polymorphic variations were found in the non-coding regions and analyzed in two groups of children stratified by low and high pro-NT levels. Finally, a most promising variant rs2234762 was found located close to the interesting regulatory region at 5’ termini. The authors have convincingly shown that carriers of such variant rs2234762 have many advantages compared to wild type: reduced pro-NT level, lower LDL-cholesterol level, less obesity, less pronounced insulin resistance (HOMA-IR). Multiple linear regression analysis validates the data obtained adjusting for potential confounders (sex, age, BMI etc). The only factor reducing the significance of the work is that only Caucasian children were analyzed and the effect of rs2234762 variant might depend on the ethnic group. The authors agree that current study should be considered as a pilot one and genetic analysis of more wide population groups is necessary to confirm the significance of rs2234762 variant. In any case the data obtained is interesting and should be published after correction of the slip on line 340 “confiurmation” should be changed to confirmation.

We do thank the reviewer for the stimulating comments. We agree that our observations warrant confirmation in different ethnic populations and in larger cohorts.

The typo on line 340 has been corrected.

Please see more comments in the attachment.

  1. What is the main question addressed by the research?
    The article by Sentinelli F. et al completes a series of works available online or even made by the authors themselves devoted to the role of plasma pro-NT, a precursor of neurotensin, in development of cardiovascular pathologies, obesity, glucose tolerance, insulin resistance and diabetes. Here the authors decided to use genetic analysis to evaluate the possible genetic regulation of plasma pro-NT level to improve metabolic parameters of patients.
  2. Do you consider the topic original or relevant in the field? Does it address a specific gap in the field? The topic is really original. Most of genetic analysis of neurotensin gene (NTS) and its polymorphism is associated with the function of neurotensin as neuropeptide in central nervous system (for instance in legs tremor, alcohol dependency etc) while genetic regulation of its activity in gut as a regulator of fat ingestion remains a gap. With regard to other NTS polymorphic alleles found previously the authors mention rs1800832 which according to studies by Riezzo et al and Russo et al [31-33] is associated with the reduced levels of pro-NT in plasma but current work by Sentinelli F.et al did not find the difference and rs1800832 was equally distributed in low- and high- pro-NT groups.
  3. What does it add to the subject area compared with other published material? They add possible mechanism of genetic regulation of pro-NT level.
  4. What specific improvements should the authors consider regarding the methodology? What further controls should be considered? As I mentioned in my review it is necessary to confirm the effect of rs2234762 variant on the principally different ethnic group.
  5. Are the conclusions consistent with the evidence and arguments presented and do they address the main question posed? Yes.
  6. Are the references appropriate? Yes.
  7. Please include any additional comments on the tables and figures.

As additional comments I’d recommend to explain the abbreviation GG-, CC-, genotypes which is not so common and why C>G corresponds to rs2234762 carrier. It is not so principle as the reader could guess (table 2) that CG and GG genotypes carry rs2234762 variant in hetero and homozygote state while CC (wild type) does not contain it but why it is named CC- etc remains obscure.

The rs2234762 polymorphism consists in a substitution of the nucleotide C (cytosine) to the G (guanine) nucleotide indicated as C>G change. Therefore, the abbreviation CC genotype indicates subjects carrying two copies of the major allele C, and the GG genotype specifies subjects with two copies of the minor allele G. Those subjects with both C and G alleles are indicated as CG heterozygotes. All the explanations regarding the genotype abbreviations have been added to the legend of Table 2 (Page 5, lines 9-10).

Besides on line 13 genotype allele frequencies are approved not to deviate from Hardy-Weinberg with p>0.05. Why p>0.05 is valid should be explained.

Genotypes and allele frequencies did not deviate from Hardy–Weinberg equilibrium taking in account our value of c2 test = 0.65 that is below the critical value of 3.84 with 1 degree of freedom (p > 0.05)”. This elucidation has been added at page 3 last line and page 4 lines 1-2.

I have no comments to tables 1and 3 as statistics is made in proper mode and conclusions correspond to the contents of the tables. Among abbreviations to table 2 I did not find FBI though it is present in Materials and Methods Section.

Thank you for pointing out the error in Table 2 regarding the abbreviation of the insulin parameter. Now, it has been corrected.

For table 4 (logistic regression analysis) more comments are necessary. The authors should explain what cut-off points were used for variables Age and BMI to transform them to binary predictors that were used in logistic regression.

In the logistic regression analysis showed in table 4, we transformed the HOMA-IR parameter as a dichotomous variable considering the median value and included it in the test as a dependent variable. The parameters age and BMI, selected as independent variables, were included in the test as continuous variables.

As requested by the referee we added more comments regarding the logistic regression analysis showed in table 4 at page 6 lines 3-7.

Noteworthy, the NTS rs2234762 variant was again associated with less pronounced insulin resistance, i.e. at the logistic regression analysis, the NTS rs2234762 variant associated with lower HOMA-IR distribution at the 6.5 year follow-up (HOMA-IR index below the median value 2.84) after adjusting for age, sex and BMI with OR: 0.46 (95%C.I. 0.216 – 0.983) (Table 4).”